# Decision-making in acute viral bronchiolitis: A universal guideline and a publication gap

Jochen Meyburg[1], Markus Ries [2]*

1 Department of General Pediatrics and Pediatric Intensive Care, Center of Pediatric and Adolescent Medicine, University Hospital Heidelberg, Heidelberg, Germany, 2 Pediatric Neurology and Metabolic Medicine, Center of Pediatric and Adolescent Medicine, University Hospital Heidelberg, Heidelberg, Germany

* markus.ries@uni-heidelberg.de

## Abstract

### Background

Acute viral bronchiolitis is very common in infants and children up to 2 years. Some patients develop serious respiratory symptoms and need to be hospitalized. In 2014, the American Academy of Pediatrics (AAP) published a guideline on acute bronchiolitis which has gained global acceptance. We hypothesized that a publication gap, which is increasingly perceived in clinical medicine, might have also affected these universal recommendations.

### Methods

We determined the proportion of published and unpublished studies registered at Clinical-Trials.gov that were marked as completed by October 1st 2018. The major trial and literature databases were used to search for publications. In addition, the study investigators were contacted directly.

### Results

Of the 69 registered studies on the treatment of acute viral bronchiolitis, only 50 (72%) have been published by November 2019. Published trials contained data from n = 9403 patients, whereas n = 4687 patients were enrolled in unpublished trials. Median time to publication was 20 months, and only 8 of 50 trials were published within 12 months after completion. Only 40% of the clinical trials that were completed after the release of the AAP guideline were subsequently published as compared to 80% before 2014.

### Conclusion

There is a significant publication gap regarding therapy of acute viral bronchiolitis that may have influenced certain recommendations of the AAP guideline. In turn, recommendations of the guideline might have discouraged investigators to publish their results after its release.

**Data Availability Statement:** All relevant data are within the manuscript and its supporting information file.

**Funding:** The authors received no specific funding for this work.

**Competing interests:** The authors have declared that no competing interests exist.

## Introduction

Viral bronchiolitis is a very common cause of outpatient visits in children younger than 2 years and accounts for 16% of hospitalizations in this age group [1]. Although several viruses may cause bronchiolitis, respiratory syncytial virus (RSV) is responsible for about 80% of infections, especially in infants [2]. Whereas the infection has a benign and self-limiting course in the majority of affected children, it can be a serious and potentially life-threatening disease in some patients with complex underlying diseases such as congenital heart disease or bronchopulmonary dysplasia [3].

Unfortunately, a causative treatment of acute viral bronchiolitis does not exist due to its special pathophysiology. Infection of the terminal bronchiolar epithelial cells causes edema, excessive formation of mucus, and eventually necrosis of bronchiolar epithelia. Hence, therapy is restricted to symptomatic measures until the respiratory epithelia have regenerated. Such supportive care may include a variety of medications, including chest physiotherapy and breathing aids, and different caregivers have been using different approaches in the past. In 2014, a landmark paper by the American Association of Pediatrics was published which has been regarded a universal clinical practice guideline since [4]. Surprising to many, the AAP guideline does not support the routine use of commonly used therapies such as chest physiotherapy, bronchodilators (e.g. albuterol, or salbutamol), epinephrine inhalation, corticosteroids, or antibiotics. Recommendations were only made for administration of nebulized hypertonic saline and nasogastral or intravenous fluids, both in hospitalized children only.

The AAP guideline was developed by a group of very renowned experts based on published study results. However, it is increasingly noted that the results of many clinical trials are not reported in a timely manner or not reported at all. Such selective reporting of study results, known as publication bias or publication gap has been observed in a various fields of pediatrics [5–8]. Given the discrepancy between the few treatment options recommended by the AAP guideline and the widespread use of various other therapeutic approaches, we were wondering whether the actual decision making might be influenced by such a publication gap.

## Methods

### Identification of clinical trials

To identify registered clinical trials on bronchiolitis reported as completed, the ClinicalTrials.gov database provided by the U.S. National Library of Medicine was assessed. Search criteria were: keyword "bronchiolitis" with the query selection parameters "completed studies" and "child (0–17 years)". Close of database was October 1$^{st}$ 2019. Data were downloaded for further analysis.

### Search for publications of completed trials

To identify publications related to the registered and completed trials, ClinicalTrials.gov, PubMed and Google Scholar were searched for NCT number, study title, principal investigator, study sponsor and keywords generated from the study title. If no respective publication was found, the principal investigators were contacted by email and/or social networks (ResearchGate and/or facebook) and asked to provide information whether the study was published in a source not covered by PubMed or Google Scholar.

### Data analysis

The STROBE criteria (STrengthening the Reporting of OBservational studies in Epidemiology) were applied for design and analysis of this study [9]. Data were analyzed for age and

number of participants, gender, study type, study design, condition, intervention, availability of study results, completion date, publication date, sponsor and country of sponsor. Trials were categorized into 11 groups according to their main research topic. Time to publication was calculated as the difference in months between study completion date and publication date. Missing data were not imputed. All statistical analyses were performed in SPSS 20 (IBM Corporations, Armonk, New York) using standard methods for descriptive statistics.

## Results

### Publication status of studies

We identified a total of 124 studies that were reported as completed in the ClinicalTrials.gov database. Thirty of these studies were not related to viral bronchiolitis, the majority of which studies on bronchiolitis obliterans following lung transplantation. Of the remaining 94 clinical trials, 71 investigated therapeutic interventions in acute viral bronchiolitis. Two of these studies, both unpublished, were completed less than one year before close of the database. Because the U.S. Food and Drug Administration (FDA) allows a time frame of one year between completion and publication of the study as specified in the FDA Drug Administration Amendments Act (FDAAA) [10], these two studies were excluded from the analysis. Of the remaining 69 studies, 50 were published and 19 were unpublished (S1 Table). Among them were two clinical trials that had been reported as conference papers only. One of these abstracts gave sufficient data that supported a therapeutic recommendation. We therefore regarded the study as published. The other abstract, however, only contained vague information on the study results, therefore we considered the study unpublished. All but one principal investigator of the unpublished studies could be contacted by email or social networks. Of these 18 authors, three replied and confirmed that the study results had not been published yet (Fig 1). Publication rates considerably varied between different countries of the sponsor (Table 1) and main topics of the investigations (Table 2).

The numbers of published and unpublished studies for each year of study completion (2003–2018) is shown in Fig 2. While 80% of the registered clinical trials that had been completed up to 2014 have been subsequently published, it is noticeable that almost 40% of the studies completed after the release of the AAP guideline still remain unpublished.

### Patient numbers

All studies involved both genders. Published trials contained data from n = 9403 patients, whereas n = 4687 patients were enrolled in unpublished trials. Median size of published trials was 93 (IQR 48–175), range 12–1636 individuals, whereas median size of unpublished trials was 94 (IQR 60–146), range 33–2580 participants. Fig 3 shows that in some years, the number of patients enrolled in unpublished studies significantly exceeded those in published studies.

### Time to publication

Median time to publication was 20 months (IQR 13–31), range 2 to 58 months. No trend could be identified that either older or more recent studies were published faster, and only 8 of 50 trials were published within 12 months after completion as warranted by the FDAAA (Fig 4).

## Discussion

Few other guidelines have gained quasi universal global acceptance as the 2014 AAP guideline on bronchiolitis [4]. Therefore, our analysis of the registered clinical trials on the treatment of

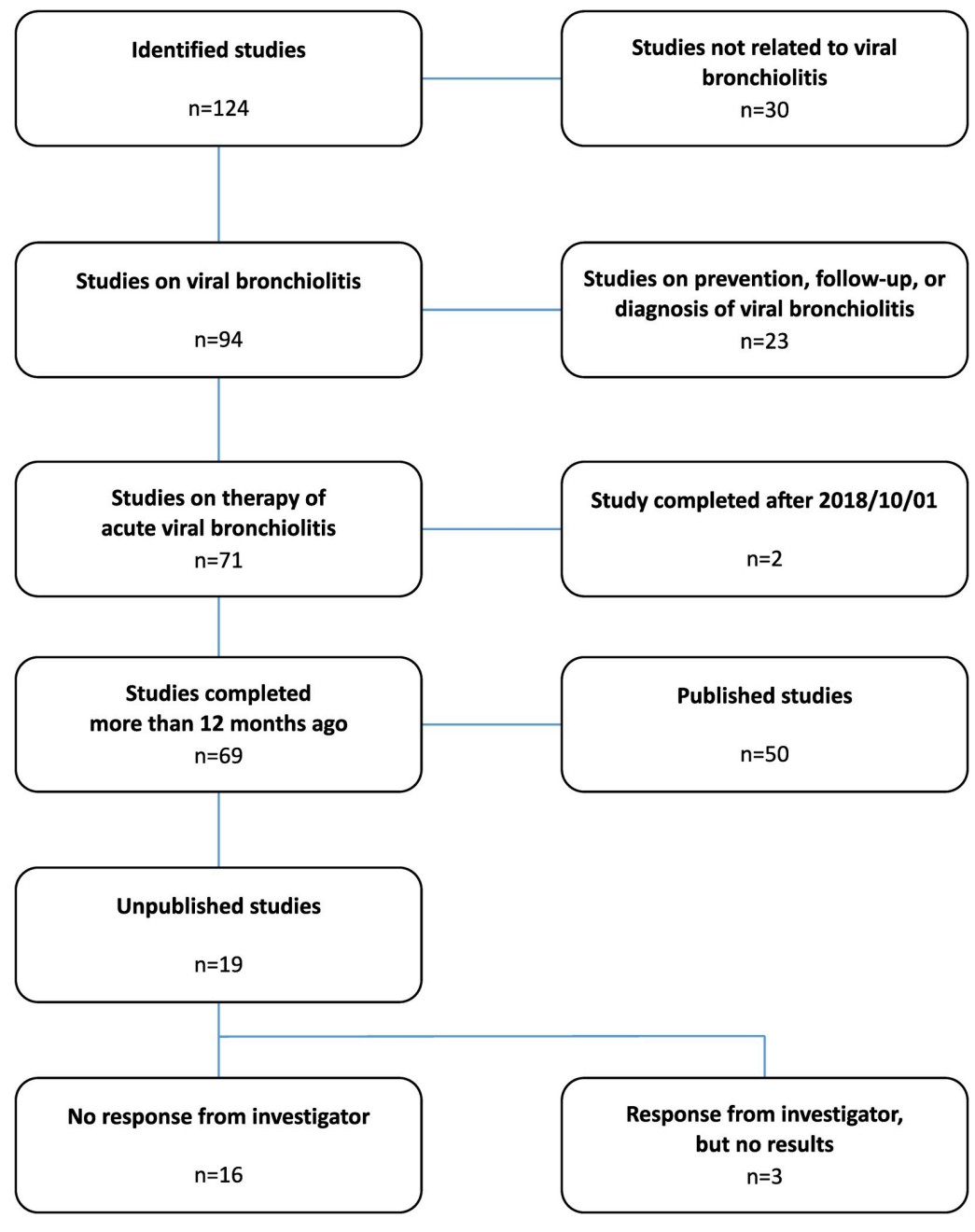

**Fig 1. Flowsheet: Details of the study selection process.**

acute viral bronchiolitis had two key questions: (1) Could a publication gap have influenced the guideline's recommendations and (2) did the guideline affect the publication status of subsequent clinical trials?

For the first part of our analysis, only clinical trials the results of which could have been available prior to completion of the AAP guideline were taken into account. Few therapeutic options have been recommended in the guideline: oxygen (given an oxygen saturation below 90%), adequate fluid intake, and inhalation with hypertonic saline. No unpublished studies on supplemental oxygen or fluid intake were identified. However, two studies on the use of

**Table 1. Countries of the sponsor.**

| Countries | Published studies (n) | Unpublished studies (n) |
|---|---|---|
| Mexico | 0 | 2 |
| Argentina, Bangladesh, Chile, India, Iran, Lebanon | 0 | 1 |
| Australia, Belgium, China, Denmark, Finland, Nepal, Netherlands, Switzerland, Thailand, United Kingdom | 1 | 0 |
| Egypt | 1 | 1 |
| Spain | 2 | 1 |
| Brazil, Italy, Turkey | 2 | 0 |
| Canada | 3 | 1 |
| France | 5 | 4 |
| Israel | 5 | 1 |
| Qatar | 5 | 0 |
| United States | 13 | 3 |

Published (n = 50) and unpublished (n = 19) completed therapeutic studies on viral bronchiolitis by country.

hypertonic saline have been completed, but not published before the AAP guideline was released. The guideline gives a weak recommendation to administer nebulized hypertonic saline to infants and children hospitalized for bronchiolitis, but does not recommend its use in the emergency department (ED). One of the unpublished studies compared hospitalization rates of infants presenting to the ED with bronchiolitis during the year of use of nebulized hypertonic saline versus the two previous years when nebulized hypertonic saline was not used in a single center in France (NCT01460524). This study alone enrolled 2580 patients, which is about 10% more than the sum of all patients (2294) from 14 studies analyzed in the AAP guideline. So the results of this trial might well have affected the recommendations for the use of hypertonic saline in the ED. The other unpublished clinical trial (NCT01238848) studied the effects of hypertonic saline versus normal saline in combination with albuterol in 82 children already hospitalized for moderate bronchiolitis. It is unlikely that this study would have had an impact on the AAP recommendations.

**Table 2. Main study topics.**

| Issue | Overall number of studies | Number and percentage of published studies | Number of patients enrolled in unpublished studies |
|---|---|---|---|
| Hypertonic Saline | 23 | 16 (70%) | 3154 |
| Respiratory Support (Oxygen, HFNC, CPAP) | 10 | 7 (70%) | 380 |
| Other drugs | 7 | 4 (57%) | 440 |
| Physiotherapy | 7 | 6 (86%) | 204 |
| Epinephrine | 6 | 3 (50%) | 239 |
| Montelukast | 6 | 4 (67%) | 287 |
| Steroids | 6 | 4 (67%) | 214 |
| Gases (NO, Helium) | 4 | 3 (75%) | 69 |
| Furosemide | 2 | 2 (100%) | 0 |
| Magnesium | 2 | 2 (100%) | 0 |
| Organisation/Isolation | 2 | 2 (100%) | 2 |

Publication status of studies registered as completed on ClinicalTrials.gov involving children with bronchiolitis. HFNC: High flow nasal cannula, CPAP: continuous positive airway pressure, NO: nitric oxide.

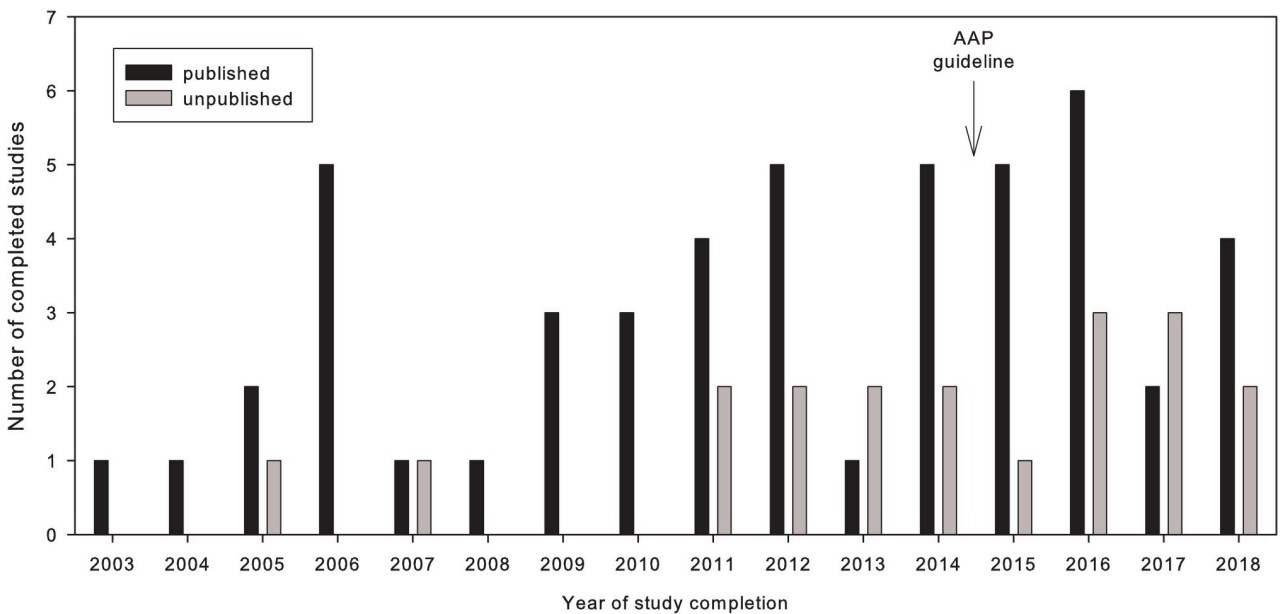

**Fig 2. Distribution of published (n = 50) and unpublished (n = 19) trials by year of completion.** Close of database was October 1st 2019.

The AAP guideline explicitly advises against various other therapeutic measures: bronchodilators (albuterol or salbutamol), antibiotics, epinephrine, steroids, and chest physiotherapy. In our analysis, we did not find unpublished studies on the use of bronchodilators or antibiotics. One unpublished clinical trial with 82 participants (NCT00435994) investigated the production of VEGF from nasal washing after inhalation with epinephrine in infants with bronchiolitis and healthy controls, respectively. Thus, published results of this study would not have affected the guideline. We identified one unpublished trial with 94 patients on the use of

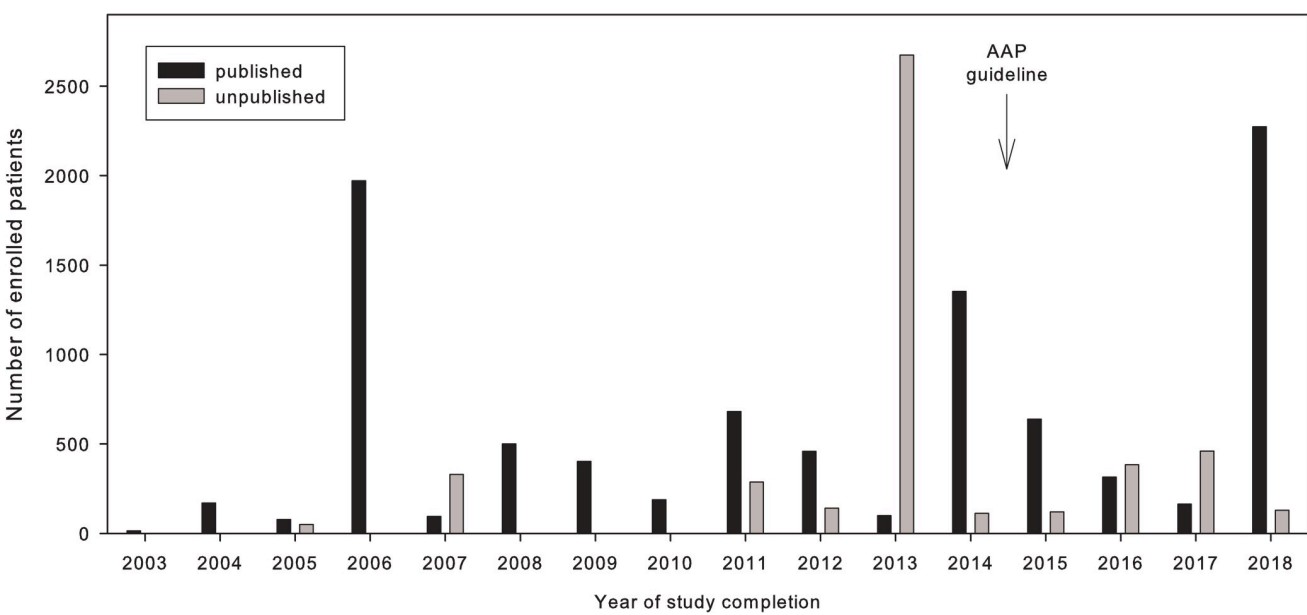

**Fig 3. Distribution of patient count stratified by publication status and year.** Close of database was October 1st 2019.

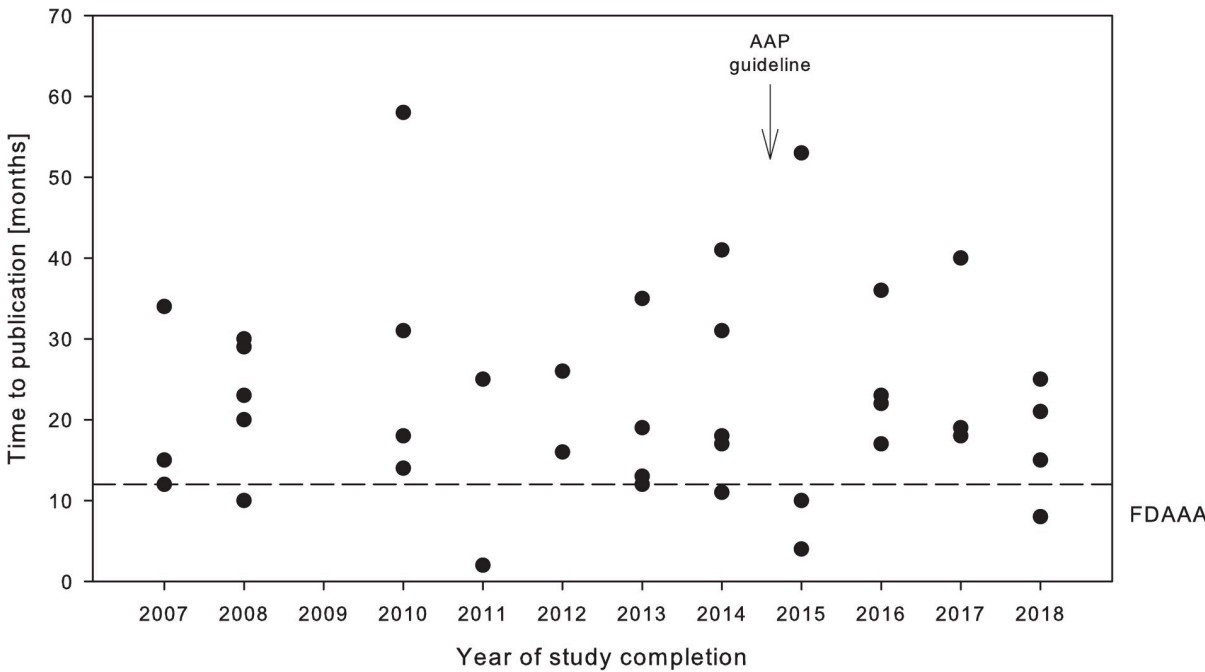

**Fig 4. Time to publication (time between completion of the trial and publication of results) in months by year of completion.** Close of database was October 1st 2019.

steroids in acute bronchiolitis (NCT02571517). In this study, children that had been hospitalized for moderate to severe bronchiolitis received intravenous methylprednisolone and/or oral prednisolone for seven days. Several clinical endpoints (disease severity scores, chest X-rays, admission to the PICU, need for mechanical ventilation) were defined as primary and secondary outcomes on day 7. Interestingly, the AAP guideline focuses on hospital admission rates regarding the use of steroids, but does not give evidence for a lack of efficiency in children already hospitalized for bronchiolitis. Thus, the results of this unpublished study might have had an impact on the official recommendations. Several of the recommendations in the guideline are supported by low-quality evidence, e.g. the recommendation not to administer supplemental oxygen if $SpO_2$ >90% (Expert Opinion; Evidence quality D). Many infants with bronchiolitis upon discharge from the hospital have lower basal $SpO_2$ than healthy control infants and an elevated oxygen desaturation index during sleep [11]. The pathophysiological and clinical consequences of mild nocturnal hypoxemia and especially of the intermittent type are unknown, therefore, further research is needed.

There are also therapies for bronchiolitis on which clinical trials are listed in the register, but that are not addressed in the guideline. Montelukast is mentioned in Appendix 1 as a MedLine search term, but its use is not discussed in the text. The Clinical trials database lists a total of six studies on montelukast in bronchiolitis, two of which only investigated the pharmacokinetics. Of the remaining four clinical trials, two have been published. A small study with 53 participants found no benefit of montelukast in patients with acute disease [12]. A second multicenter study with 1125 participants focused on post-RSV bronchiolitis during an observation periods of 24 weeks [13]. Again, beneficial effects of montelukast were not observed. On the other hand, we identified two unpublished randomized controlled trials on the use of montelukast in acute bronchiolitis, both with significant numbers of participants (NCT01370187: n = 146; NCT00863317: n = 141). It can be speculated that the AAP would have advised against

montelukast in the guideline if the results of the two unpublished studies had come to the same conclusion as the published ones.

Finally, we analyzed the database to find out whether the AAP guideline might have affected the publication status of clinical trials in children with acute bronchiolitis. It was impressive that the rate of unpublished studies doubled after the release of the guideline. Among the unpublished studies completed after 2014, four were on the use of nebulized hypertonic saline (NCT02538458, NCT02233985, NCT03143231, NCT03614273). It is likely that given the guideline's at least weak recommendation to use it in hospitalized patients, there is a publication bias regarding positive results. In other words, the mere confirmation of the official recommendations may not have been worth the effort of publication. The same may hold true for two other unpublished studies on the combined use of hypertonic saline, dexamethasone, and epinephrine (NCT01834820) and chest physiotherapy, respectively (NCT02853838). On the other hand, there are possible therapies that might be addressed in a revision of the guideline in the near future, such as the use of high flow nasal cannula (HFNC). Since there are several clinical studies that found positive effects of HFNC in acute bronchiolitis [14–16], maybe this will motivate the principal investigators to publish the results of three recent studies that are yet unpublished (NCT02791711, NCT01498094, NCT02856165).

This study has several limitations. We did not investigate other clinical trials databases (e.g., EU-CTR or the German Clinical Trials Register), because ClinicalTrials.gov is generally considered the largest and most important clinical trial registry. Only registered clinical trials could be analyzed because the existence of non-registered trials was not transparent to us. In order to avoid a clinical trial being erroneously classified as unpublished, we conducted a semantic literature search in PubMed and GoogleScholar and we contacted investigators and sponsors. We assumed that the data submitted to ClinicalTrials.gov were accurate and complete as mandated by the FDAAA [10]. We did not compare whether the pre-specified statistical analysis plan and the pre-specified research questions of the study were consistent with the published reports, because this information was not in the public domain for all studies.

In conclusion, publication bias or publication gap is an ubiquitous issue in clinical science. Increasingly, study results are not reported in a timely manner, or not published at all. The therapy of acute viral bronchiolitis is a major task for pediatricians during the winter season, and it is supported by a highly appreciated guideline that is followed worldwide. But even this guideline might have been affected by a publication gap and in turn might have discouraged investigators to publish their results after its release.

## Supporting information

**S1 Table. List of published (n = 50) and unpublished (n = 19) trials.** Close of database was October 1st 2019.
(DOCX)

## Author Contributions

**Conceptualization:** Jochen Meyburg, Markus Ries.

**Data curation:** Jochen Meyburg.

**Formal analysis:** Jochen Meyburg.

**Investigation:** Jochen Meyburg.

**Methodology:** Jochen Meyburg, Markus Ries.

**Supervision:** Markus Ries.

**Validation:** Jochen Meyburg.

**Writing – original draft:** Jochen Meyburg.

**Writing – review & editing:** Markus Ries.

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
