## [Decision Letter · Decision Letter 0]

22 Jul 2020

PONE-D-20-14816

Decision-making in acute viral bronchiolitis a universal guideline and a publication gap

PLOS ONE

Dear Dr. Ries,

Thank you for submitting your manuscript to PLOS ONE. After careful consideration, we feel that it has merit but does not fully meet PLOS ONE’s publication criteria as it currently stands. Therefore, we invite you to submit a revised version of the manuscript that addresses the points raised during the review process.

We look forward to receiving your revised manuscript.

Kind regards,

Ivan D. Florez

Academic Editor

PLOS ONE

Additional Editor Comments:

Your manuscript has been reviewed by two experts in the field, and they have found some points that need to be addressed before this manuscript is considered for publication. Please go through the reviewers' comments and consider addressing these points, and prepare a revised version.

Journal Requirements:

2. Please include a caption for figure 4.

Reviewers' comments:

Reviewer's Responses to Questions

**Comments to the Author**

1. Is the manuscript technically sound, and do the data support the conclusions?

Reviewer #1: Yes

Reviewer #2: Yes

2. Has the statistical analysis been performed appropriately and rigorously? 

Reviewer #1: Yes

Reviewer #2: Yes

3. Have the authors made all data underlying the findings in their manuscript fully available?

Reviewer #1: Yes

Reviewer #2: Yes

4. Is the manuscript presented in an intelligible fashion and written in standard English?

Reviewer #1: Yes

Reviewer #2: Yes

5. Review Comments to the Author

Reviewer #1: Please check in results: patients numbers: Subjects mean age, comorbilities, because this is an important information for the treatment issue.

Line 135/136 You can improve the idea

line 143, please check it. It is better to write 20 months

Reviewer #2: This is a well-written manuscript demonstrating the publication gap regarding treatment of viral bronchiolitis following publication of the relevant 2014 AAP Clinical Practice Guideline. Based on recommendations included in the Clinical Practice Guideline, most traditional treatment interventions are discouraged and discharge to home is encouraged if the infant does not show any signs of bacterial infection, has adequate po intake and there is only mild or no hypoxemia. Several of the recommendations are supported by low-quality evidence.

One characteristic example that the authors need to mention in their Discussion is the recommendation not to administer supplemental oxygen if SpO2 >90% (Expert Opinion; Evidence quality D). Many infants with bronchiolitis upon discharge from the hospital have lower basal SpO2 than healthy control infants and an elevated oxygen desaturation index during sleep (Kaditis et al. Infants with viral bronchiolitis demonstrate two distinct patterns of nocturnal oxyhaemoglobin desaturation. Acta Paediatrica 2015; 104: e106-111). The pathophysiological and clinical consequences of mild nocturnal hypoxemia and especially of the intermittent type are unknown.

The consequence of the publication gap in the field is that many clinicians use interventions that are not recommended by the AAP Guidleine like inhaled bronchodilators and supplemental oxygen.

6. PLOS authors have the option to publish the peer review history of their article (what does this mean?). If published, this will include your full peer review and any attached files.

Reviewer #1: No

Reviewer #2: No

---

## [Author Response · Author response to Decision Letter 0]

31 Jul 2020

Response to Reviewers

Decision-making in acute viral bronchiolitis: a universal guideline and a publication gap

We thank the reviewers for their thoughtful comments that were very helpful to further strengthen the manuscript. We are addressing the comments – numbered in consecutive order - in the following section point by point:

Journal Requirements:

Comment 1: Please ensure that your manuscript meets PLOS ONE's style

requirements, including those for file naming. 

Answer 1: done.

Comment 2: Please include a caption for figure 4.

Answer 2: The caption was included in the results section (lines 148-149).

Comment 3: Please include captions for your Supporting Information files at the

end of your manuscript, and update any in-text citations to match

accordingly. Please see our Supporting Information guidelines for more

information: http://journals.plos.org/plosone/s/supporting-information

Answer 3: The caption was included at the end of the manuscript. In-text citations were checked and are appropriate. The file was named accordingly.

Reviewer #1:

Comment 4: Please check in results: patients numbers: Subjects mean age, comorbilities, because this is an important information for the treatment issue.

Answer 4: We thank the reviewer for this important comment. The patient numbers for published and unpublished studies, respectively, are given in the results section, lines 134-137. For three of the unpublished studies, preliminary results are posted on the ClinicalTrials.gov website. However, these data are very scarce. Mean age of the participants (4.5 ± 3.8 months; 8.2 ± 5.5 months; 11 ± 6 months) was not different from the mean age of the published studies and it was consistent with the typical population of pediatric patients with acute viral bronchiolitis. No information about comorbidities are available for any of the unpublished studies. We agree with the reviewers that patient numbers, patient age, and comorbidities may be substantial confounders important for clinical practice. However, from the available published data we are not able to provide further insight into this issue.

Comment 5: Line 135/136 You can improve the idea

Answer 5: We agree that this phrase was difficult to understand and added a word for clarification. 

Comment 6: line 143, please check it. It is better to write 20 months

Answer 6: We agree with the reviewer and edited the respective phrase accordingly (line 143).

Reviewer #2:

Comment 7: This is a well-written manuscript demonstrating the publication gap regarding treatment of viral bronchiolitis following publication of the relevant 2014 AAP Clinical Practice Guideline. Based on recommendations included in the Clinical Practice Guideline, most traditional treatment interventions are discouraged and discharge to home is encouraged if the infant does not show any signs of bacterial infection, has adequate po intake and there is only mild or no hypoxemia. Several of the recommendations are supported by low-quality evidence.

One characteristic example that the authors need to mention in their Discussion is the recommendation not to administer supplemental oxygen if SpO2 >90% (Expert Opinion; Evidence quality D). Many infants with bronchiolitis upon discharge from the hospital have lower basal SpO2 than healthy control infants and an elevated oxygen desaturation index during sleep (Kaditis et al. Infants with viral bronchiolitis demonstrate two distinct patterns of nocturnal oxyhaemoglobin desaturation. Acta Paediatrica 2015; 104: e106-111). The pathophysiological and clinical consequences of mild nocturnal hypoxemia and especially of the intermittent type are unknown.

The consequence of the publication gap in the field is that many clinicians use interventions that are not recommended by the AAP Guidleine like inhaled bronchodilators and supplemental oxygen.

Answer 7: We thank the reviewer for this comment. We agree and added a paragraph and a reference into the discussion section.

---

## [Decision Letter · Decision Letter 1]

4 Aug 2020

Decision-making in acute viral bronchiolitis: a universal guideline and a publication gap

PONE-D-20-14816R1

Dear Dr. Ries,

We’re pleased to inform you that your manuscript has been judged scientifically suitable for publication and will be formally accepted for publication once it meets all outstanding technical requirements.

Kind regards,

Ivan D. Florez

Academic Editor

PLOS ONE

Additional Editor Comments (optional):

All the comments from the reviewers have been addressed.

Reviewers' comments:

Reviewer's Responses to Questions

**Comments to the Author**

1. If the authors have adequately addressed your comments raised in a previous round of review and you feel that this manuscript is now acceptable for publication, you may indicate that here to bypass the “Comments to the Author” section, enter your conflict of interest statement in the “Confidential to Editor” section, and submit your "Accept" recommendation.

Reviewer #1: All comments have been addressed

Reviewer #2: All comments have been addressed

2. Is the manuscript technically sound, and do the data support the conclusions?

Reviewer #1: Yes

Reviewer #2: Yes

3. Has the statistical analysis been performed appropriately and rigorously? 

Reviewer #1: Yes

Reviewer #2: Yes

4. Have the authors made all data underlying the findings in their manuscript fully available?

Reviewer #1: Yes

Reviewer #2: Yes

5. Is the manuscript presented in an intelligible fashion and written in standard English?

Reviewer #1: Yes

Reviewer #2: Yes

6. Review Comments to the Author

Reviewer #1: (No Response)

Reviewer #2: (No Response)

7. PLOS authors have the option to publish the peer review history of their article (what does this mean?). If published, this will include your full peer review and any attached files.

Reviewer #1: No

Reviewer #2: No

---

## [Editor Report · Acceptance letter]

6 Aug 2020

PONE-D-20-14816R1 

Decision-making in acute viral bronchiolitis:
a universal guideline and a publication gap 

Dear Dr. Ries:

I'm pleased to inform you that your manuscript has been deemed suitable for publication in PLOS ONE. Congratulations! Your manuscript is now with our production department. 

Kind regards, 

on behalf of

Dr. Ivan D. Florez 

Academic Editor

PLOS ONE